Confidence intervals for ratio of means of delta-lognormal distributions based on left-censored data with application to rainfall data in Thailand

Thangjai Warisa 1
Niwitpong Sa-Aat sa-aat.n@sci.kmutnb.ac.th 2
1 Department of Statistics, Ramkhamhaeng University , Bangkok , Thailand
2 Department of Applied Statistics, King Mongkut’s University of Technology North Bangk , Bangkok , Thailand
Mahmood Haider
Electronic publication date: 2023 Nov 9
Publication date: 2023
Volume: 11
Electronic Location ID: e16397
Received 2023 Sep 1; Accepted 2023 Oct 12
Copyright: ©2023 Thangjai and Niwitpong
Copyright year: 2023
Copyright holder: Thangjai and Niwitpong
License: This is an open access article distributed under the terms of the Creative Commons Attribution License, which permits unrestricted use, distribution, reproduction and adaptation in any medium and for any purpose provided that it is properly attributed. For attribution, the original author(s), title, publication source (PeerJ) and either DOI or URL of the article must be cited.
License URL: https://creativecommons.org/licenses/by/4.0/

Keywords: Bayesian credible interval, Delta-lognormal distribution, Left-censored data, Ratio, Rainfall

Funding: Office of the Permanent Secretary, Ministry of Higher Education, Science, Research and Innovation (OPS MHESI), Thailand Science Research and Innovation (TSRI) and Ramkhamhaeng University RGNS 65-178 This work (grant no. RGNS 65-178) was supported by Office of the Permanent Secretary, Ministry of Higher Education, Science, Research and Innovation (OPS MHESI), Thailand Science Research and Innovation (TSRI) and Ramkhamhaeng University. The funders had no role in study design, data collection and analysis, decision to publish, or preparation of the manuscript.

==============================
Thailand is a country that is prone to both floods and droughts, and these natural disasters have significant impacts on the country’s people, economy, and environment. Estimating rainfall is an important part of flood and drought prevention. Rainfall data typically contains both zero and positive observations, and the distribution of rainfall often follows the delta-lognormal distribution. However, it is important to note that rainfall data can be censored, meaning that some values may be missing or truncated. The interval estimator for the ratio of means will be useful when comparing the means of two samples. The purpose of this article was to compare the performance of several approaches for statistically analyzing left-censored data. The performance of the confidence intervals was evaluated using the coverage probability and average length, which were assessed through Monte Carlo simulation. The approaches examined included several variations of the generalized confidence interval, the Bayesian, the parametric bootstrap, and the method of variance estimates recovery approaches. For (ξ1, ξ2) = (0.10,0.10), simulations showed that the Bayesian approach would be a suitable choice for constructing the credible interval for the ratio of means of delta-lognormal distributions based on left-censored data. For (ξ1, ξ2) = (0.10,0.25), the parametric bootstrap approach was a strong alternative for constructing the confidence interval. However, the generalized confidence interval approach can be considered to construct the confidence when the sample sizes are increase. Practical applications demonstrating the use of these techniques on rainfall data showed that the confidence interval based on the generalized confidence interval approach covered the ratio of population means and had the smallest length. The proposed approaches’ effectiveness was illustrated using daily rainfall datasets from the provinces of Chiang Rai and Chiang Mai in Thailand.

Introduction

Floods occur in Thailand primarily during the monsoon season, which typically lasts from May to October. During this time, heavy rainfall can cause rivers to overflow and inundate low-lying areas, leading to widespread flooding. In some cases, flash floods can also occur, particularly in urban areas with poor drainage systems. The impacts of floods in Thailand can include damage to crops and infrastructure, loss of property and livelihoods, and loss of life. On the other hand, droughts are more common during the dry season, which typically lasts from November to April. During this time, rainfall is limited, and water resources such as rivers, reservoirs, and groundwater become depleted. Droughts can have significant impacts on agriculture, which is a major sector of Thailand’s economy. They can also lead to water shortages for households and industries, as well as damage to the environment. Rainfall estimation is an essential tool for addressing these challenges and preventing floods and droughts in Thailand. By accurately estimating rainfall patterns, authorities can better prepare for and mitigate the impacts of these natural disasters, helping to protect the country’s people, economy, and environment. The analysis and modeling of rainfall patterns play a crucial role in various fields, such as hydrology, agriculture, and climate science. Thailand experiences a wide range of rainfall patterns due to its geographic location, varying topography, and monsoonal climate. The characteristics of rainfall data fit various distributions, such as the normal distribution, log-normal distribution, gamma distribution, Weibull distribution, inverse Gaussian distribution, and delta-lognormal distribution. Several researchers have indicated that the delta-lognormal distribution is a suitable model for rainfall data in Thailand; see Krstanovic & Singh (1992), Singh & Rajagopal (1986) and Singh & Singh (1987).

The delta-lognormal distribution is a statistical distribution commonly used to model continuous positive data, such as rainfall, crop yields, and stock prices. It is a two-part distribution that contains both zero and positive values. The number of zero values follows a binomial distribution, while the positive values follow a log-normal distribution. The log-normal distribution is formed by taking the exponential of a normal distribution. The delta-lognormal distribution is often used in hydrology and other fields to model variables that exhibit high variability and are bounded by zero. It has been found to be an effective model for a wide range of applications, particularly for modeling rainfall data. Several researchers have studied confidence intervals for functions of the delta-lognormal distribution. Thangjai et al. (2023) proposed the confidence interval estimation for the ratio of the percentiles of two delta-lognormal distributions. Additionally, given that rainfall data is often censored, it is crucial to estimate the functions of the delta-lognormal distribution based on left-censored data for obtaining accurate statistics. Thangjai & Niwitpong (2023) estimated the confidence intervals for mean and difference between means of delta-lognormal distributions based on left-censored data with application to rainfall data.

The difference of parameters refers to the subtraction or comparison of two parameters or coefficients in a statistical model. This is typically done to quantify the difference in the impact or effect of two variables on an outcome. In rainfall data analysis, the difference of parameters is used to assess the difference in the effects of two variables. For instance, the difference between the coefficients of rainfall associated with different geographical regions is computed to determine if one region experiences significantly more or less rainfall than the other. The ratio of parameters refers to the division or comparison of two parameters or coefficients in a statistical model. Parameters can represent various aspects of a model, such as the coefficients of variables in a regression model, the proportions in a probability distribution, or the odds ratios in logistic regression. In the context of rainfall data analysis, the ratio of parameters is used to compare the effects of different variables on rainfall. For example, the ratio of the coefficients of temperature and humidity in a regression model is used to determine how each of these factors contributes to changes in rainfall. Moreover, the difference of means is likely to be minor when both means are small, and such a minor difference can lead to an inability to draw powerful or definite conclusions. Therefore, the ratio of means is often considered more accurate than the difference of means. The ratio of means is the ratio of two means and is used in many fields. For instance, in bioequivalence, the ratio of means is used to compare the mean of the test drug and the mean of the reference drug. In epidemiology, the ratio of means is used to compare the particulate matter with a diameter of less than 2.5 µm (PM2.5) level averages of two areas. In climate sciences, the ratio of means is used to compare the daily rainfall averages of two areas. Confidence intervals for the ratio of means have been constructed in many research studies, such as those by Chen & Zhou (2006a).

In statistics, the information in a sample is used to make inferences about an unknown parameter. The inference methods are hypothesis testing and estimation (Casella & Berger, 2002). Estimations have a point estimation and an interval estimation. Estimation is of interest in many fields. For example, in the environment, Luo, Shen & Xu (2022) studied the modeling and estimation of system reliability under dynamic operating environments and lifetime ordering constraint. They used the maximum likelihood method for point estimation while proposing generalized inference methods for interval estimation. In industry, Zhang et al. (2022) studied the problem of reliability estimation for a parallel system when one stress variable is involved, referred to as the multicomponent stress–strength model.

The construction of confidence intervals is a crucial aspect of statistical inference, and many researchers have proposed various approaches for constructing such intervals. The generalized confidence interval (GCI) approach uses the concepts of the generalized pivotal quantity (GPQ) to construct the confidence interval. Chen & Zhou (2006a) presented the GCI estimate for the ratio and the difference between the means of log-normal distributions. It gave a highly accurate coverage rate and fairly low bias, especially for small sample sizes. Tian & Wu (2007) proposed the GCI approach for inferences on the common mean of log-normal distributions. Ye, Ma & Wang (2010) proposed inferences on the common mean of several inverse Gaussian populations using the GCI approach.

The bootstrap approach relies heavily on computer simulations. Traditionally, standard errors have been calculated using well-known formulae, often based on assumptions that are not satisfied or only approximately satisfied. In some cases, it may not even be known if the assumptions hold or not. In essence, the bootstrap approach relies on resampling with replacement from the given sample and calculating the required statistic from these repeated samples. The values of the statistic from the repeated sampling can then be used to generate standard errors and confidence intervals for the statistic (Dunn, 2001). Thangjai et al. (2023) constructed the confidence interval for the ratio of the percentiles of two delta-lognormal distributions based on the parametric bootstrap approach. Moreover, Altunkaynak & Gamgam (2019) proposed the bootstrap confidence intervals for the coefficient of quartile variation.

The method of variance estimates recovery (MOVER) approach utilizes the initial confidence interval of a single parameter of interest to construct the final confidence interval. Zou & Donner (2008) constructed the confidence limits about effect measures using the MOVER approach. Zou, Taleban & Hao (2009) proposed the MOVER approach to estimate the confidence interval for log-normal distribution.

Statistics can be divided into two different techniques: the classical approach and the Bayesian approach. The classical approach includes techniques such as the GCI, parametric bootstrap, and MOVER. In this approach, the parameter of interest is unknown but fixed. In contrast, the Bayesian approach considers the parameter of interest as a quantity, and its variation is described by the prior distribution. There are many reasons why a researcher may prefer to use Bayesian estimation over classical estimation. The main reason for choosing the Bayesian approach is that the models are often too complex for traditional methods to handle. It is important to note that, regardless of the reasons for implementing the Bayesian approach, conducting a sensitivity analysis of priors is always crucial and should be included (Depaoli, Winter & Visser, 2020). The impact of the priors is highly dependent on model complexity, and it is crucial to thoroughly examine their influence on the final model estimates. The Bayesian approach can effectively capture the features of posteriors of the parameters of interest by combining information from data and priors. However, obtaining closed forms for each marginal posterior in Bayesian analysis is a challenging task. Markov chain Monte Carlo (MCMC) can be employed to obtain posterior samples from a set of Markov chains with respect to the parameters of interest and nuisance parameters. The iteration is terminated when all chains are stable and well-mixed. MCMC has been well-developed and widely utilized for complex models, allowing parameter estimation through posterior samples generated from a collection of Markov chains (Zhou et al., 2023). Thangjai et al. (2023) proposed the credible interval estimation for the ratio of the percentiles of two delta-lognormal distributions using the Bayesian approach. Furthermore, Thangjai & Niwitpong (2023) constructed the Bayesian credible interval for mean and difference between means of delta-lognormal distributions based on left-censored data. Moreover, the Bayesian approach is also widely used for uncertainty quantification (Zhuang, Xu & Wang, 2023). Aizpurua et al. (2022) studied how uncertainty quantification was incorporated into machine health prognostics through the Bayesian approach.

The rest of this article is organized as follows. ‘Materials & Methods’ describes the four approaches used to construct confidence intervals for the ratio of means of delta-lognormal distributions based on left-censored data. ‘Results’ presents the performance of the proposed approaches using simulation studies. ‘Empirical Application’ shows a real data example. ‘Discussion’ provides a discussion. Finally, ‘Conclusions’ presents concluding remarks.

Materials & Methods

Suppose Z1=Z11,Z12,…,Z1n1 is random sample from delta-lognormal distribution with parameters mean μ1, variance σ12, and probability of obtaining a zero observation δ1. Similary, let Z2 = (Z21, Z22, …, Z2n2) be random sample following the delta-lognormal distribution with parameters mean μ2, variance σ22, and probability of obtaining a zero observation δ2. The delta-lognormal distribution is combination of zero and positive values. This distribution contains a binomial distribution and a log-normal distribution. This is because the zero values follow the binomial distribution and the positive values follow the log-normal distribution. The mean of delta-lognormal distribution for Z1 and Z2 are given by (1) γ1=1−δ1expμ1+12σ12

and (2) γ2=1−δ2expμ2+12σ22.

Suppose X1 and X2 are nonnegative random variables drawn from Z1 and Z2, respectively. In other words, X1 = (X11, X12, …, X1n1) and X2 = (X21, X22, …, X2n2) follow the log-normal distributions so that Y1 = log(X1) and Y2 = log(X2) follow the normal distributions. Suppose that X ¯1 and X ¯2 are the means of X1 and X2, respectively. Moreover, suppose that SX12 and SX22 are the variances of X1 and X2, respectively. Let log(ξ1) be censoring point value. Let n1(1) be the number of observations less than or equal to some censoring point log(ξ1) and let n1(2) be the number of observations greater than some censoring point log(ξ1). Let Y1=Y11,Y12,…,Y1n12 be the observations above log(ξ1). The mean and variance of Y1 are given by (3) Y ¯1=1n12 ∑j=1n12Y1j

and (4) S12=1n12 ∑j=1n12Y1j−Y ¯12.

Suppose that ϕ and Φ are the density function and the distribution function of the standard normal distribution. According to Krishnamoorthy, Mallick & Mathew (2011), the maximum likelihood estimators of μ1 and σ12 are given by (5) μ ˆ1=Y ¯1−ψh1,a1Y ¯1− logξ1

and (6) σ ˆ12=S12+ψh1,a1Y ¯1− logξ12,

where h1=n11n1

a1=logξ1−μ1σ1

Wa1=ϕa11−Φa1

Vh1,a1=h1W−a11−h1

and ψh1,a1=Vh1,a1Vh1,a1−a1.

The mean of censored log-normal distribution is (7) θ1= expμ1+12σ12.

The estimator of the mean of censored log-normal distribution is (8) θ ˆ1= expμ ˆ1+12σ ˆ12.

Similarly, let log(ξ2) be censoring point value. Let n2(1) be the number of observations less than or equal to log(ξ2) and let n2(2) be the number of observations greater than some censoring point log(ξ2). Let Y2=Y21,Y22,…,Y2n22 be the observations above log(ξ2). The mean and variance of Y2 are given by (9) Y ¯2=1n22 ∑j=1n22Y2j

and (10) S22=1n22 ∑j=1n22Y2j−Y ¯22.

The maximum likelihood estimators of μ2 and σ22 are given by (11) μ ˆ2=Y ¯2−ψh2,a2Y ¯2− logξ2

and (12) σ ˆ22=S22+ψh2,a2Y ¯2− logξ22,

where h2=n21n2

a2=logξ2−μ2σ2

Wa2=ϕa21−Φa2

Vh2,a2=h2W−a21−h2

and ψh2,a2=Vh2,a2Vh2,a2−a2.

The mean of censored log-normal distribution is (13) θ2= expμ2+12σ22.

The estimator of the mean of censored log-normal distribution is (14) θ ˆ2= expμ ˆ2+12σ ˆ22.

Therefore, the estimator of the ratio of means of censored log-normal distributions is given by (15) θ ˆ=θ ˆ1θ ˆ2,

where θ ˆ1 and θ ˆ2 are defined in Eqs. (8) and (14), respectively.

The estimator of the ratio of means of censored log-normal distributions, defined in Eq. (15), is used to construct the confidence intervals for the ratio of means of delta-lognormal distributions based on left-censored data. Here, four newly proposed approaches are applied to construct the confidence intervals. Next, the computation of the GCI, Bayesian, parametric bootstrap, and MOVER approaches is explained.

Generalized confidence interval approach

The generalized pivotal quantity (GPQ) is used to construct the GCI which is defined in Weerahandi (1993). According to Krishnamoorthy, Mallick & Mathew (2011), the GPQs for μ1, σ1, and θ1 are defined by (16) Rμ1=μ ˆ1−μ ˆ1∗σ ˆ1∗σ ˆ1

(17) Rσ1=σ ˆ1σ ˆ1∗

and (18) Rθ1= expRμ1+12Rσ12,

where μ ˆ1∗ and σ ˆ1∗ are the maximum likelihood estimators based on a censored sample from standard normal distribution.

Similarly, the GPQs for μ2, σ2, and θ2 are defined by (19) Rμ2=μ ˆ2−μ ˆ2∗σ ˆ2∗σ ˆ2

(20) Rσ2=σ ˆ2σ ˆ2∗

and (21) Rθ2= expRμ2+12Rσ22,

where μ ˆ2∗ and σ ˆ2∗ are the maximum likelihood estimators based on a censored sample from standard normal distribution.

The GPQ for the difference between means of delta-lognormal distributions based on left-censored data was used as previously described in Thangjai & Niwitpong (2023). In this article, the GPQ for the ratio of means of delta-lognormal distributions based on left-censored data is given by (22) Rθ=Rθ1Rθ2,

where Rθ1 and Rθ2 are defined in Eqs. (18) and (21), respectively.

Therefore, the 100(1 − α)% two-sided confidence interval for the ratio of means of delta-lognormal distributions based on left-censored data using the GCI approach is given by (23) CIGCI=LGCI,UGCI=Rθα/2,Rθ1−α/2,

where Rθ(α/2) and Rθ(1 − α/2) denote the 100(α/2)-th and 100(1 − α/2)-th percentiles of Rθ, respectively.

The Algorithm 1 is used to construct the GCI for the ratio of means of delta-lognormal distributions based on left-censored data.

Algorithm 1.

Step 1: Generate sample from the standard normal distribution and compute μ ˆ1∗, μ ˆ2∗, σ ˆ1∗, and σ ˆ2∗

Step 2: Compute Rμ1, Rσ1, and Rθ1 from Eqs. (16), (17), and (18), respectively.

Step 3: Compute Rμ2, Rσ2, and Rθ2 from Eqs. (19), (20), and (21), respectively.

Step 4: Compute Rθ from Eq. (22)

Step 5: Repeat step 1 - step 4, a total times and obtain an array of Rθ’s

Step 6: Compute LGCI and UGCI

Bayesian approach

The Bayesian approach offers a framework for updating beliefs and making predictions using new evidence or data. It is rooted in Bayes’ theorem, which combines prior probability and likelihood to calculate the posterior probability. The prior distribution represents uncertainty about parameters before observing data. In this article, we employed the Jeffreys Independence prior. According to Thangjai & Niwitpong (2023), the posterior distributions of σ12, μ1, and θ1.BS are defined by (24) σ12|y1∼IGn12−12,n12−1s122

(25) μ1|σ12,y1∼Ny ¯1,σ12n12

and (26) θ1.BS= expμ1+12σ12,

where y ¯1 is observed value of Y ¯1 defined in Eq. (3) and s12 is the observed value of S12 defined in Eq. (4).

The posterior distributions of σ22, μ2, and θ2.BS are defined by (27) σ22|y2∼IGn22−12,n22−1s222

(28) μ2|σ22,y2∼Ny ¯2,σ22n22

and (29) θ2.BS= expμ2+12σ22,

where y ¯2 is observed value of Y ¯2 defined in Eq. (9) and s22 is the observed value of S22 defined in Eq. (10).

Thangjai & Niwitpong (2023) proposed the posterior distribution of the difference between means of delta-lognormal distributions based on left-censored data. Therefore, the posterior distribution of the ratio of means of delta-lognormal distributions based on left-censored data is given by (30) θBS=θ1.BSθ2.BS,

where θ1.BS and θ2.BS are defined in Eqs. (26) and (29), respectively.

Therefore, the 100(1 − α)% two-sided credible interval for the ratio of means of delta-lognormal distributions based on left-censored data using the Bayesian approach is given by (31) CIθ.BS=Lθ.BS,Uθ.BS,

where Lθ.BS and Uθ.BS denote the lower and upper limits of the shortest 100(1 − α)% highest posterior density interval of θBS, respectively.

The Algorithm 2 is used to construct the Bayesian credible interval for the ratio of means of delta-lognormal distributions based on left-censored data.

Algorithm 2.

Step 1: Compute σ12|y1, μ1|σ12,y1, and θ1.BS from Eqs. (24), (25), and (26), respectively.

Step 2: Compute σ22|y2, μ2|σ22,y2, and θ2.BS from Eqs. (27), (28), and (29), respectively.

Step 3: Compute θBS from Eq. (30)

Step 4: Repeat step 1 - step 3, a total m times and obtain an array of θBS’s

Step 5: Compute Lθ.BS and Uθ.BS

Parametric bootstrap approach

Let Y1∗=Y11∗,Y12∗,…,Y1n12∗ be the observations above log(ξ1). The mean and variance of Y1∗ are given by (32) Y ¯1∗=1n12 ∑j=1n12Y1j∗

and (33) S12∗=1n12 ∑j=1n12Y1j∗−Y ¯1∗2.

The estimator of the mean of delta-lognormal distribution based on left-censored data is (34) θ ˆ1∗= expY ¯1∗+12S12∗,

where Y ¯1∗ and S12∗ are defined in Eqs. (32) and (33), respectively.

Let Y2∗=Y21∗,Y22∗,…,Y2n22∗ be the observations above log(ξ2). The mean and variance of Y2∗ are given by (35) Y ¯2∗=1n22 ∑j=1n22Y2j∗

and (36) S22∗=1n22 ∑j=1n22Y2j∗−Y ¯2∗2.

The estimator of the mean of delta-lognormal distribution based on left-censored data is (37) θ ˆ2∗= expY ¯2∗+12S22∗,

where Y ¯2∗ and S22∗ are defined in Eqs. (35) and (36), respectively.

The estimator of the difference between means of delta-lognormal distributions based on left-censored data was proposed as previously described in Thangjai & Niwitpong (2023). According to Thangjai & Niwitpong (2023), the estimator of the ratio of means of delta-lognormal distributions based on left-censored data is given by (38) θ ˆ∗=θ ˆ1∗θ ˆ2∗,

where θ ˆ1∗ and θ ˆ2∗ are defined in Eqs. (34) and (37), respectively.

The lower and upper limits of the confidence interval for the ratio of means of delta-lognormal distributions based on left-censored data are given by (39) LPB=θ ˆ∗¯−z1−α/2sdθ ˆ∗

and (40) UPB=θ ˆ∗¯+z1−α/2sdθ ˆ∗,

where θ ˆ∗¯ is the mean of θ ˆ∗, sdθ ˆ∗ is the standard deviation of θ ˆ∗, and z1−α/2 is the 100(1 − α/2)-th percentile of the standard normal distribution.

Therefore, the 100(1 − α)% two-sided confidence interval for the ratio of means of delta-lognormal distributions based on left-censored data using the parametric bootstrap approach is given by (41) CIPB=LPB,UPB,

where LPB and UPB are defined in Eqs. (39) and (40), respectively.

The Algorithm 3 is used to construct the parametric bootstrap confidence interval for the ratio of means of delta-lognormal distributions based on left-censored data.

Algorithm 3.

Step 1: Parametric bootstrapping assumes that the data comes from a known distribution with unknown parameters. We estimate these parameters from the data and then use the estimated distributions to simulate the samples. Generate y1∗=y11∗,y12∗,…,y1n12∗ from normal distribution with μ ˆ1 and σ ˆ12 and generate y2∗=y21∗,y22∗,…,y2n22∗ from normal distribution with μ ˆ2 and σ ˆ22

Step 2: Compute y ¯1∗ from Eq. (32), s12∗ from Eq. (33), and θ ˆ1∗ from Eq. (34)

Step 3: Compute y ¯2∗ from Eq. (35), s22∗ from Eq. (36), and θ ˆ2∗ from Eq. (37)

Step 4: Compute θ ˆ∗ from Eq. (38)

Step 5: Repeat step 1 - step 4, a total m times and obtain an array of θ ˆ∗’s

Step 6: Compute LPB and UPB from Eqs. (39) and (40)

Method of variance estimates recovery approach

According to Maneerat, Niwitpong & Niwitpong (2020), the lower and upper limits of the confidence interval for μ1 are defined by (42) lμ1=μ ˆ1−z1−α/2n12−1σ ˆ12n12χn12−12

and (43) uμ1=μ ˆ1+z1−α/2n12−1σ ˆ12n12χn12−12,

where z1−α/2 is the 100(1 − α/2)-th percentile of the standard normal distribution, χn12−12 is the chi-squared distribution with n1(2) − 1 degrees of freedom, and μ ˆ1 and σ ˆ12 are defined in Eqs. (5) and (6), respectively.

Similarly, the lower and upper limits of the confidence interval for μ2 are given by (44) lμ2=μ ˆ2−z1−α/2n22−1σ ˆ22n22χn22−12

and (45) uμ2=μ ˆ2+z1−α/2n22−1σ ˆ22n22χn22−12,

where z1−α/2 is the 100(1 − α/2)-th percentile of the standard normal distribution, χn22−12 is the chi-squared distribution with n2(2) − 1 degrees of freedom, and μ ˆ2 and σ ˆ22 are defined in Eqs. (11) and (12), respectively.

Following Maneerat, Niwitpong & Niwitpong (2020), the lower and upper limits of the confidence interval for σ12 are defined by (46) lσ12=n12−1σ ˆ12χ1−α/2,n12−12

and (47) uσ12=n12−1σ ˆ12χα/2,n12−12,

where χ1−α/2,n12−12 and χα/2,n12−12 are the 100(1 − α/2)-th and 100(α/2)-th percentiles of the chi-squared distribution with n1(2) − 1 degrees of freedom, and σ ˆ12 is defined in Eq. (6).

The lower and upper limits of the confidence interval for σ22 are defined by (48) lσ22=n22−1σ ˆ22χ1−α/2,n22−12

and (49) uσ22=n22−1σ ˆ22χα/2,n22−12,

where χ1−α/2,n22−12 and χα/2,n22−12 are the 100(1 − α/2)-th and 100(α/2)-th percentiles of the chi-squared distribution with n2(2) − 1 degrees of freedom, and σ ˆ22 is defined in Eq (12).

Applying the concept of Donner & Zou (1993), the lower and upper limits of confidence interval for θ1= expμ1+12σ12 are given by (50) lθ1= expμ ˆ1+12σ ˆ12−μ ˆ1−lμ12+12σ ˆ12−12lσ122

and (51) uθ1= expμ ˆ1+12σ ˆ12+uμ1−μ ˆ12+12uσ12−12σ ˆ122,

where μ ˆ1, σ ˆ12, lμ1, uμ1, lσ12, and uσ12 are defined in Eqs. (5), (6), (42), (43), (46), and (47), respectively.

Similarly, the lower and upper limits of confidence interval for θ2= expμ2+12σ22 are given by (52) lθ2= expμ ˆ2+12σ ˆ22−μ ˆ2−lμ22+12σ ˆ22−12lσ222

and (53) uθ2= expμ ˆ2+12σ ˆ22+uμ2−μ ˆ22+12uσ22−12σ ˆ222,

where μ ˆ2, σ ˆ22, lμ2, uμ2, lσ22, and uσ22 are defined in Eqs. (11), (12), (44), (45), (48), (49), respectively.

Using the concept of Donner & Zou (2012) and Thangjai & Niwitpong (2023), the lower and upper limits of confidence interval for θ=θ1θ2 are given by (54) LMOVER=θ ˆ1θ ˆ2−θ ˆ1θ ˆ22−lθ1uθ22θ ˆ1−lθ12θ ˆ2−uθ2uθ22θ ˆ2−uθ2

and (55) UMOVER=θ ˆ1θ ˆ2+θ ˆ1θ ˆ22−uθ1lθ22θ ˆ1−uθ12θ ˆ2−lθ2lθ22θ ˆ2−lθ2,

where θ ˆ1, θ ˆ2, lθ1, uθ1, lθ2, and uθ2 are defined in Eqs. (8), (14), (50), (51), (52), and (53), respectively.

Therefore, the 100(1 − α)% two-sided confidence interval for the ratio of means of delta-lognormal distributions based on left-censored data using the MOVER approach is given by (56) CIMOVER=LMOVER,UMOVER,

where LMOVER and UMOVER are defined in Eqs. (54) and (55), respectively.

Results

In this section, we conducted simulation studies to evaluate the performance of the proposed confidence interval, which was constructed using four different approaches. We calculated the coverage probability and average length using R software. The criteria for choosing the best performing confidence interval were a coverage probability greater than or equal to 0.95 and the shortest average length for each tested scenario. For each generated data set, we used R code to compute the confidence intervals based on the GCI, Bayesian, parametric bootstrap, and MOVER approaches (Thangjai & Niwitpong, 2023), with M = 5,000 runs for each and m = 2,500 runs for the GCI, Bayesian, and parametric bootstrap approaches. Following Owen & De Rouen (1980), the standardized sensitivity for many air contaminants is typically around 0.25. Although 0.25 was determined to be the most appropriate value for ξ when examining the censoring techniques, additional runs were performed using ξ = 0.10, allowing for the examination of results for the delta distribution with different values of μ and σ. We generated random sample sizes of (n1, n2) = (20,20), (30,30), (20,30), (50,50), (30,50), (100,100), and (50,100) with specific parameters, as described in Table 1.

Table 1 Values selected for the population means, population standard deviations, probabilities of obtaining zero observation, and censoring points.

Run number	(μ1, μ2)	(σ1, σ2)	(δ1, δ2)	(ξ1, ξ2)	
0	(0.00,0.00)	(0.30,0.30)	(0.10,0.10)	No censoring	
1	(0.00,0.00)	(1.00,1.00)	(0.10,0.10)	(0.10,0.10)	
2	(0.00,0.00)	(1.00,1.00)	(0.10,0.10)	(0.10,0.25)	
3	(0.00,0.00)	(1.00,1.00)	(0.10,0.25)	(0.10,0.10)	
4	(0.00,0.00)	(1.00,1.00)	(0.10,0.25)	(0.10,0.25)	
5	(0.00,0.00)	(1.00,2.00)	(0.10,0.10)	(0.10,0.10)	
6	(0.00,0.00)	(1.00,2.00)	(0.10,0.10)	(0.10,0.25)	
7	(0.00,0.00)	(1.00,2.00)	(0.10,0.25)	(0.10,0.10)	
8	(0.00,0.00)	(1.00,2.00)	(0.10,0.25)	(0.10,0.25)	

The Algorithm 4 is used to construct the confidence intervals for the ratio of means of delta-lognormal distributions based on left-censored data, and then the coverage probability and average length of the confidence intervals are computed.

Algorithm 4.

Step 1: Generate z1 from delta-lognormal distribution with parameters μ1, σ1, and δ1 and set x1 from log-normal distribution with parameters μ1 and σ1

Step 2: Generate z2 from delta-lognormal distribution with parameters μ2, σ2, and δ2 and set x2 from log-normal distribution with parameters μ2 and σ2

Step 3: Compute y1 = log(x1) and select y1 > log(ξ1)

Step 4: Compute y2 = log(x2) and select y2 > log(ξ2)

Step 5: Compute n1(1), n1(2), n2(1), n2(2), μ ˆ1, μ ˆ2, σ ˆ1, σ ˆ2, θ ˆ1, θ ˆ2, and θ ˆ

Step 6: Construct the confidence intervals CIGCI, CIBS, CIPB, and CIMOVER

Step 7: If L ⩽ θ ⩽ U, set p = 1; else set p = 0

Step 8: Compute U − L

Step 9: Repeat step 1 - step 8, a total M times

Step 10: Compute mean of p defined by the coverage probability

Step 11: Compute mean of U − L defined by the average length

Table 2 The coverage probabilities (CPs) and average lengths (ALs) of 95% two-sided confidence intervals for the ratio of means of delta-lognormal distributions based on left-censored data.

(n1, n2)	Run number	CP (AL)	
		CI GCI	CI BS	CI PB	CI MOVER	
(20, 20)	0	0.9442	0.9570	0.9376	0.9480	
		(0.4111)	(0.4326)	(0.3913)	(0.4358)	
	1	0.9822	0.9812	0.9396	0.9870	
		(4.2060)	(3.5736)	(2.1539)	(5.1344)	
	2	0.9796	0.9850	0.9488	0.9850	
		(4.6143)	(3.8363)	(2.2088)	(6.4682)	
	3	0.9862	0.9858	0.9258	0.9948	
		(4.5750)	(4.0626)	(2.4027)	(6.2972)	
	4	0.9828	0.9904	0.9574	0.9898	
		(5.0102)	(4.2274)	(2.4848)	(6.2869)	
	5	0.9566	0.9718	0.9668	0.9774	
		(1.7387)	(1.5506)	(1.5333)	(2.4622)	
	6	0.9540	0.9768	0.9642	0.9772	
		(1.7741)	(1.6403)	(1.5060)	(2.6641)	
	7	0.9476	0.9718	0.9578	0.9864	
		(1.8618)	(1.7981)	(1.8586)	(3.0510)	
	8	0.9342	0.9754	0.9614	0.9848	
		(1.9484)	(1.9493)	(1.8062)	(5.8979)	
(30,30)	0	0.9344	0.9506	0.9334	0.9460	
		(0.3275)	(0.3440)	(0.3233)	(0.3468)	
	1	0.9816	0.9824	0.9420	0.9868	
		(2.6356)	(2.3787)	(1.6508)	(2.8968)	
	2	0.9808	0.9878	0.9582	0.9832	
		(2.8724)	(2.4416)	(1.6901)	(2.9633)	
	3	0.9880	0.9878	0.9366	0.9952	
		(2.8720)	(2.7300)	(1.8001)	(3.4266)	
	4	0.9798	0.9924	0.9614	0.9880	
		(3.2285)	(2.8937)	(1.9375)	(3.5624)	
	5	0.9540	0.9678	0.9612	0.9766	
		(1.0608)	(0.9934)	(1.0431)	(1.2940)	
	6	0.9520	0.9760	0.9630	0.9786	
		(1.0498)	(1.0423)	(1.0376)	(1.3833)	
	7	0.9340	0.9622	0.9432	0.9810	
		(1.1444)	(1.1641)	(1.2322)	(1.5948)	
	8	0.9378	0.9784	0.9628	0.9816	
		(1.1568)	(1.2536)	(1.2359)	(1.7320)	
(20,30)	0	0.9428	0.9568	0.9406	0.9508	
		(0.3748)	(0.3932)	(0.3589)	(0.3964)	
	1	0.9814	0.9858	0.9328	0.9874	
		(3.9451)	(3.4056)	(1.9245)	(4.9194)	
	2	0.9778	0.9866	0.9418	0.9814	
		(4.3365)	(3.5499)	(1.9858)	(5.1065)	
	3	0.9856	0.9814	0.9080	0.9928	
		(4.0345)	(3.5815)	(2.0299)	(5.1957)	
	4	0.9820	0.9938	0.9616	0.9886	
		(4.7817)	(4.0717)	(2.2738)	(5.8989)	
	5	0.9610	0.9722	0.9580	0.9768	
		(1.3996)	(1.2598)	(1.1147)	(1.9116)	
	6	0.9572	0.9774	0.9670	0.9824	
		(1.3685)	(1.3347)	(1.1302)	(2.0536)	
	7	0.9372	0.9638	0.9416	0.9818	
		(1.4648)	(1.4277)	(1.3012)	(2.2402)	
	8	0.9382	0.9826	0.9658	0.9856	
		(1.5085)	(1.5732)	(1.3543)	(2.5036)	
(50,50)	0	0.9374	0.9496	0.9422	0.9488	
		(0.2475)	(0.2593)	(0.2508)	(0.2609)	
	1	0.9824	0.9806	0.9460	0.9868	
		(1.7057)	(1.6105)	(1.1956)	(1.7847)	
	2	0.9808	0.9876	0.9632	0.9826	
		(1.8963)	(1.6690)	(1.2599)	(1.8472)	
	3	0.9854	0.9842	0.9246	0.9932	
		(1.8930)	(1.9056)	(1.3053)	(2.1481)	
	4	0.9754	0.9924	0.9600	0.9810	
		(2.1639)	(2.0206)	(1.4505)	(2.2564)	
	5	0.9516	0.9660	0.9578	0.9750	
		(0.6564)	(0.6582)	(0.6677)	(0.7660)	
	6	0.9446	0.9722	0.9636	0.9764	
		(0.6627)	(0.7121)	(0.7045)	(0.8355)	
	7	0.9254	0.9540	0.9288	0.9776	
		(0.6808)	(0.7374)	(0.7530)	(0.9003)	
	8	0.9276	0.9760	0.9594	0.9834	
		(0.6866)	(0.8167)	(0.8121)	(0.9944)	
(30,50)	0	0.9376	0.9510	0.9390	0.9464	
		(0.2918)	(0.3055)	(0.2894)	(0.3074)	
	1	0.9852	0.9866	0.9382	0.9902	
		(2.3965)	(2.1589)	(1.4367)	(2.5955)	
	2	0.9836	0.9900	0.9534	0.9844	
		(2.6092)	(2.2311)	(1.5015)	(2.6781)	
	3	0.9846	0.9820	0.9158	0.9922	
		(2.4721)	(2.3535)	(1.5021)	(2.8636)	
	4	0.9818	0.9932	0.9682	0.9888	
		(2.8244)	(2.5417)	(1.6759)	(3.0589)	
	5	0.9550	0.9676	0.9498	0.9774	
		(0.8059)	(0.7777)	(0.7231)	(0.9666)	
	6	0.9412	0.9736	0.9584	0.9794	
		(0.7694)	(0.8141)	(0.7389)	(1.0140)	
	7	0.9362	0.9606	0.9284	0.9804	
		(0.8027)	(0.8387)	(0.7959)	(1.0721)	
	8	0.9284	0.9782	0.9594	0.9816	
		(0.8330)	(0.9602)	(0.8796)	(1.2372)	
(100,100)	0	0.9400	0.9530	0.9512	0.9520	
		(0.1726)	(0.1807)	(0.1784)	(0.1820)	
	1	0.9852	0.9858	0.9530	0.9888	
		(1.0970)	(1.0656)	(0.8239)	(1.1218)	
	2	0.9696	0.9832	0.9512	0.9730	
		(1.2069)	(1.0866)	(0.8579)	(1.1443)	
	3	0.9832	0.9862	0.9238	0.9934	
		(1.2068)	(1.2656)	(0.8807)	(1.3390)	
	4	0.9716	0.9872	0.9372	0.9724	
		(1.3686)	(1.3111)	(0.9788)	(1.3837)	
	5	0.9462	0.9610	0.9438	0.9734	
		(0.3998)	(0.4269)	(0.4103)	(0.4614)	
	6	0.9156	0.9706	0.9596	0.9758	
		(0.3991)	(0.4655)	(0.4399)	(0.5040)	
	7	0.9014	0.9382	0.9036	0.9712	
		(0.4029)	(0.4698)	(0.4404)	(0.5244)	
	8	0.8904	0.9720	0.9538	0.9804	
		(0.4092)	(0.5367)	(0.5033)	(0.5946)	
(50,100)	0	0.9422	0.9530	0.9482	0.9510	
		(0.2137)	(0.2237)	(0.2177)	(0.2250)	
	1	0.9846	0.9838	0.9432	0.9890	
		(1.5013)	(1.4103)	(1.0234)	(1.5517)	
	2	0.9760	0.9846	0.9564	0.9804	
		(1.6523)	(1.4719)	(1.0816)	(1.6180)	
	3	0.9842	0.9848	0.9204	0.9932	
		(1.5717)	(1.5690)	(1.0646)	(1.7336)	
	4	0.9786	0.9930	0.9642	0.9808	
		(1.8058)	(1.6931)	(1.2063)	(1.8667)	
	5	0.9508	0.9608	0.9426	0.9732	
		(0.4717)	(0.4879)	(0.4421)	(0.5458)	
	6	0.9256	0.9722	0.9534	0.9728	
		(0.4616)	(0.5328)	(0.4777)	(0.5973)	
	7	0.9098	0.9412	0.8990	0.9724	
		(0.4611)	(0.5181)	(0.4653)	(0.5922)	
	8	0.8950	0.9716	0.9502	0.9774	
		(0.4714)	(0.6041)	(0.5407)	(0.6880)	
Notes.

Bold font means the confidence interval with coverage probability greater than or equal to 0.95 and the shortest average length.

The coverage probability and average length of the confidence intervals for the ratio of means of delta-lognormal distributions based on left-censored data are presented in Table 2 and shown in Figs. 1–3. For run 0, the Bayesian approach outperforms the others in terms of both coverage probability and average length for all sample sizes. Overall, the coverage probabilities are less than or equal to the nominal confidence level of 0.95. Therefore, we used the confidence intervals for the ratio of means of delta-lognormal distributions based on left-censored data to estimate the ratio of means for datasets containing zero, positive, and censored observations. For runs 1–8, the results show that the coverage probabilities of the confidence intervals based on the GCI, Bayesian, parametric bootstrap and MOVER approaches were almost greater than the nominal confidence level of 0.95. The average lengths of the confidence interval based on the Bayesian approach were the shortest for (ξ1, ξ2) = (0.10,0.10), while the average lengths of the confidence interval based on the parametric bootstrap approach were shorter than those of the others for (ξ1, ξ2) = (0.10,0.25). The results indicate that the Bayesian approach is recommended for constructing the credible interval for the ratio of means of delta-lognormal distributions based on left-censored data for (ξ1, ξ2) = (0.10,0.10). However, the parametric bootstrap approach can be used to estimate the confidence interval for the ratio of means of delta-lognormal distributions based on left-censored data for (ξ1, ξ2) = (0.10,0.25). Moreover, the GCI approach can be used to construct the confidence interval for the ratio of means of delta-lognormal distributions based on left-censored data for run 3 and run 5 when the sample sizes are increase.

Figure 1 Comparison of the coverage probabilities and average lengths of the confidence intervals for the ratio of means of delta-lognormal distributions based on left-censored data according to sample sizes.

(A) Coverage probabilities. (B) Average lengths.

Figure 2 Comparison of the coverage probabilities and average lengths of the confidence intervals for the ratio of means of delta-lognormal distributions based on left-censored data according to probabilities of non-zero values.

(A) Coverage probabilities. (B) Average lengths.

Figure 3 Comparison of the coverage probabilities and average lengths of the confidence intervals for the ratio of means of delta-lognormal distributions based on left-censored data according to standard deviations.

(A) Coverage probabilities. (B) Average lengths.

Empirical application

The GCI, Bayesian, parametric bootstrap, and MOVER approaches discussed in ‘Materials & Methods’ can be applied to estimate the ratio of average daily rainfall datasets from Chiang Rai and Chiang Mai provinces in Thailand. Table 3 shows the daily rainfall data from June 1st to June 30th, 2022, presented by the Thai Meteorological Department. The table includes 30 observations, out of which 13 of 30 (43.33%) represent positive observed values in Chiang Rai province, and nine of 30 (30.00%) represent positive observed values in Chiang Mai province. Table 4 shows the possible distributions for the positive rainfall data applied to the minimum Akaike information criterion (AIC). Figure 4 presents the densities of the daily rainfall data in Chiang Rai and Chiang Mai provinces. Figure 5 presents the histograms of the daily rainfall data in Chiang Rai and Chiang Mai provinces. Figure 6 presents the normal QQ-plots of the log-transformed the daily rainfall data in Chiang Rai and Chiang Mai provinces. The log-transformed positive daily rainfall values of Chiang Rai and Chiang Mai provinces follow normal distributions. Therefore, the daily rainfall datasets in Chiang Rai and Chiang Mai provinces fit the delta-lognormal distributions.

Table 3 The daily rainfall data of Chiang Rai and Chiang Mai provinces.

Province	Daily rainfall data (mm)	
Chiang Rai province	0.0	0.4	0.5	0.0	0.0	
	0.0	0.1	0.0	0.2	0.0	
	0.0	0.0	0.0	3.3	21.0	
	7.9	0.2	0.0	–	2.5	
	0.0	0.0	47.5	0.5	0.0	
	10.4	0.0	0.0	9.0	–	
Chiang Mai province	0.0	0.0	0.0	0.0	41.9	
	0.4	0.0	0.0	0.1	0.0	
	0.0	0.0	0.0	0.0	1.3	
	0.0	0.2	–	–	0.0	
	0.0	0.0	23.1	0.0	0.1	
	–	0.0	–	5.4	8.7	
Notes.

Source: Thai Meteorological Department.

Table 4 The estimated AIC values for the four probability models, calculated using rainfall data in Chiang Rai and Chiang Mai provinces.

Distribution	Chiang Rai province	Chiang Mai province	
Normal	107.2557	76.5153	
Log-Normal	73.6376	51.4921	
Gamma	75.4915	52.5742	
Exponential	80.9402	58.5944	
Notes.

Bold font means the distribution with the lowest AIC value.

Figure 4 Densities of the daily rainfall data in Chiang Rai and Chiang Mai provinces.

(A) Chiang Rai Province. (B) Chiang Mai Province.

Figure 5 Histograms of the daily rainfall data in Chiang Rai and Chiang Mai provinces.

(A) Chiang Rai Province. (B) Chiang Mai Province.

Figure 6 Normal QQ-plots of the log-transformed the daily rainfall data in Chiang Rai and Chiang Mai provinces.

(A) Chiang Rai Province. (B) Chiang Mai Province.

For Chiang Rai province, the statistics are n1 = 30, n1(1) = 18, n1(2) = 12, μ ˆ1=−1.24, σ ˆ12=9.84, and θ ˆ1=39.61. For Chiang Mai province, the statistics are n2 = 30, n2(1) = 23, n2(2) = 7, μ ˆ2=−1.65, σ ˆ22=13.57, and θ ˆ2=168.88. Therefore, the ratio of means of the daily rainfall data in Chiang Rai and Chiang Mai provinces is θ ˆ=0.23. The 95% two-sided confidence intervals for the ratio of means of the daily rainfall data in Chiang Rai and Chiang Mai provinces are constructed based on GCI, Bayesian, parametric bootstrap, and MOVER approaches. For GCI approach, CIGCI = [0.0005,34.4100] with an interval length of 34.4095. For Bayesian approach, CIBS = [0.0000,291.1944] with an interval length of 291.1944. For parametric bootstrap approach, CIPB = [-54.4900,63.6449] with an interval length of 118.1349. For MOVER approach, CIMOVER = [0.0000,40960.7200] with an interval length of 40960.7200. The lower and upper limits of the 95% confidence interval correspond to the 2.50-th and 97.50-th percentiles of the average rainfall average ratio between Chiang Rai and Chiang Mai provinces. Therefore, the GCI approach has the shortest interval length. Therefore, the GCI approach is recommended for constructing the confidence intervals for the ratio of means of delta-lognormal distributions based on left-censored data. Moreover, confidence intervals for the ratio of means of delta-lognormal distributions, based on left-censored data, can be applied to environmental, meteorological, and climatological data, which often consist of positive values or exhibit right-skewed distributions, such as PM2.5 and PM10.

Discussion

Ratio of parameters focuses on the relative strength or proportion of effects, while difference of parameters emphasizes the absolute difference in the effects of two variables. Both concepts are valuable in statistical analysis, and their application in rainfall data analysis depends on the specific research question and the variables being examined. In bioassays, the ratio quantities are of potential interest. Calculating relative potency necessitates estimating the ratio of normal means. This is due to the fact that the ratio of means represents the expected values of the least squares estimates in a simple linear regression. Moreover, the problem of estimating the unoriented direction of a mean vector can lead to ratio estimation, as the direction is fully specified by the collection of all ratios of the component means (James, 1982). Therefore, the ratio of means is important. Several researchers have studied interval estimation for the ratio of means. For example, in environmental science, Zhang et al. (2021) constructed simultaneous confidence intervals for ratios of means of zero-inflated log-normal populations, with an application to rainfall data. Moreover, Singhasomboon & Piladaeng (2023) approximated the estimation of the ratio of means for log-normal distributions, applied to PM2.5 concentrations in northern Thailand. In medical science, Abdel-Karim (2015), Zhou & Tu (2020), and Chen & Zhou (2006b) estimated the ratio of means for medical costs. Furthermore, Singhasomboon, Panichkitkosolkul & Volodin (2021) constructed confidence intervals for the ratio of means to compare the survival times in months for patients who died from two cancer groups.

Thangjai et al. (2023) proposed the confidence interval estimation for the ratio of the percentiles of two delta-lognormal distributions using the Bayesian and parametric bootstrap approaches. Thangjai & Niwitpong (2023) estimated the confidence intervals for mean and difference between means of delta-lognormal distributions based on left-censored data using the GCI, Bayesian, parametric bootstrap, and MOVER approaches. In this study, we extended the GCI, Bayesian, parametric bootstrap, and MOVER approaches to construct the confidence intervals for the ratio of means of delta-lognormal distributions based on left-censored data.

We recommend using the Bayesian approach to construct the credible intervals for the ratio of means of delta-lognormal distributions based on left-censored data. This is consistent with the findings of previous studies by Thangjai et al. (2023) and Thangjai & Niwitpong (2023). Moreover, the parametric bootstrap is considered to construct the confidence interval. It is similar to Thangjai et al. (2023) and Altunkaynak & Gamgam (2019). Additionally, the GCI approach, which is similar to the method proposed by Chen & Zhou (2006a), Tian & Wu (2007), and Ye, Ma & Wang (2010), can also be used for estimating the confidence interval.

Conclusions

We constructed confidence intervals for the ratio of means of delta-lognormal distributions based on left-censored data using the GCI, Bayesian, parametric bootstrap, and MOVER approaches. The Bayesian credible intervals performed the best in terms of coverage probabilities and average lengths for (ξ1, ξ2) = (0.10,0.10). Moreover, we recommend using the GCI approach to construct the confidence interval for (ξ1, ξ2) = (0.10,0.25).

Supplemental Information

Supplemental Information 1 The daily rainfall data of Chiang Rai and Chiang Mai provinces

(A) Chiang Rai (B) Chiang Mai.

Click here for additional data file.

Additional Information and Declarations

Competing Interests

Author Contributions

Data Availability

The authors declare there are no competing interests.

Warisa Thangjai conceived and designed the experiments, performed the experiments, analyzed the data, prepared figures and/or tables, authored or reviewed drafts of the article, and approved the final draft.

Sa-Aat Niwitpong conceived and designed the experiments, authored or reviewed drafts of the article, and approved the final draft.

The following information was supplied regarding data availability:

The raw measurements are available in the Supplemental File.

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
