# Peer review of "Confidence intervals for ratio of means of delta-lognormal distributions based on left-censored data with application to rainfall data in Thailand"

_PeerJ, doi:10.7717/peerj.16397_

## Round 0.1 · original submission · Major Revisions

Please incorporate all comments of reviewers and please submit a point-to-point rebuttal letter. Particularly, the methodology and analysis sections need more attention. Please also improve the language of the paper.

**Language Note:** The Academic Editor has identified that the English language must be improved. PeerJ can provide language editing services - please contact us at copyediting@peerj.com for pricing (be sure to provide your manuscript number and title). Alternatively, you should make your own arrangements to improve the language quality and provide details in your response letter. – PeerJ Staff

Reviewer 1 ·

Basic reporting

As noted in the discussion of this manuscript, the paper proposes the CIs for the ratio of means, where it extends from Thangjai & Niwitpong, 2023 which presented the CIs for the difference of means. All use the four methods: GCI, Bayesian, parametric bootstrap and MOVER. The questions arise as follows.

- What is the difference in the meaning of ratio and difference of parameters, and if they are applied in the application, for example, rainfall data what is the difference in interpretation. This point could be provided in the motivation section and discussion.
- The authors assume the random samples Z, X and Y. However, there is no explanation about the connection of Z and X. This is the basic probability; however, it is still needed to give as the motivation. So, please give the nicely details between paragraphs 1 and 2 of Section 2.
- Similarly, in the Bayesian method, it is important to give more details on the posterior distributions. Now, (24) and (25) are provided without any information.
- In the method section, the estimator of \xi must be defined.
- For the parametric bootstrap approach, in Algorithm 3, I try to see that which is the step of re-sample with replacement and when the bootstrap difference between the bootstrap estimates and the estimate from an actual sample is computed. Furthermore, since the population standard deviation is unknown why the estimator is depended on the N(0,1).
- In my view, the algorithm does not need for the MOVER method, as it looks straightforward in computation.
- The title of Table 4 must be re-written. This is because there is no meaning for “the AIC values of the province”.
- Several papers presented CIs for the parameters in the delta lognormal distributions always use the rainfall data. Hence, the other type of data should be studied in this work, at least for the new idea for the users to use the proposed method in their applications.

Experimental design

-

Validity of the findings

-

Additional comments

-

·

Basic reporting

No comments.

Experimental design

Lines 225 and 226. I did not find M or m, and thus did not follow the suitability of the authors' simulations. Please repeat these variables here and address why/how sufficient (e.g., for the generalized confidence intervals) for the precision of the digits in Table 2.

Validity of the findings

The authors do not seem to show validity of the simulation results. Please add run number "0" (or renumber) that contains mu1=mu2=0, sigma1=sigma2=0.3 (these are characteristic for my field), delta1=delta2=0.1, and no censoring. Show that the results (e.g., GPI) are precisely correct. If necessary, set delta1=delta2=0. For bootstrapping, consider even larger sample sizes as necessary.

Additional comments

Line 89 Minor wording issue. “The bootstrap approach is a technique used to determine the accuracy of statistics.” I think that this statement is incomplete. The readers of the authors’ article will know the bootstrap. I recommend delete this one sentence, starting with the next sentence. "The bootstrap relies ..."

Reviewer 3 ·

Basic reporting

The manuscript (Confidence intervals for ratio of means of delta-lognormal distributions based on left-censored data with application to rainfall data in Thailand) is well-written and generally clear. However, there are a few instances where the language could be further improved for better clarity and precision. Additionally, some sentences or statements could be rephrased to enhance the flow and readability of the manuscript. Ensure that your citations follow a consistent format. Include the full names of authors, publication years, and complete references in the appropriate citation style.

Experimental design

no

Validity of the findings

no

Additional comments

no

---

## Round 0.2 · Major Revisions

Please incorporate all comments of the reviewer. Particularly, please focus more on the methodological issue raised by the reviewer and please submit the revision with a point-to-point rebuttal letter.

Reviewer 1 ·

Basic reporting

see below

Experimental design

see below

Validity of the findings

see below

Additional comments

Some points are not clear. Please fine below.

3. Similarly, in the Bayesian method, it is important to give more details on the posterior
distributions. Now, (24) and (25) are provided without any information.
*** From your response, why you said that "...Bayesian credible interval is the SHORTEST interval containing ..." what is the evidence? Actually, I expect that you give an explaination for obtaining the posterior distributions, as the reference in 2023 did not note about this as well.

4. In the method section, the estimator of \xi must be defined.
*** Well, it is clear that \log( \xi) is the censoring point value. However, if it is a parameter you must give the method how to estimate it. But it is not, what is it (value) in your example. This is still my question, because it is mentioned at the beginning section, but is not said later again.

5. For the parametric bootstrap approach, in Algorithm 3, I try to see that which is the step
of re-sample with replacement and when the bootstrap difference between the bootstrap
estimates and the estimate from an actual sample is computed. Furthermore, since the
population standard deviation is unknown why the estimator is depended on the N(0,1).
*** From your response (four lines), how is it different from non-parametric bootstrap. Please explain. It will be clear if in algorithm 3, first step, you give more details how to generate the data.

7. The title of Table 4 must be re-written. This is because there is no meaning for “the AIC
values of the province”.
*** Again, "The AIC values of positive rainfall data in Chiang Rai and Chiang Mai provinces" has no meaning. The AIC is used for model selection. So the title must be: The estimated AIC values for the four probability models, calculated using rainfall data in Chiang Rai and Chiang Mai provinces.

·

Basic reporting

Ok

Experimental design

Ok

Validity of the findings

Ok

Additional comments

None

---

## Round 0.3 · accepted · Accept

The paper has been improved after revisions and is accepted for publication.

Reviewer 1 ·

Basic reporting

-

Experimental design

-

Validity of the findings

-

Additional comments

-